# Extraction of Features for Time Series Classification Using Noise Injection

**DOI:** 10.3390/s24196402

**Published:** 2024-10-02

**Authors:** Gyu Il Kim, Kyungyong Chung

**Affiliations:** 1Department of Computer Science, Kyonggi University, Suwon 16227, Republic of Korea; ran6491@kyonggi.ac.kr; 2Division of AI Computer Science and Engineering, Kyonggi University, Suwon 16227, Republic of Korea

**Keywords:** time series classification, digital signal processing, data augmentation, noise injection, machine learning, deep learning

## Abstract

Time series data often display complex, time-varying patterns, which pose significant challenges for effective classification due to data variability, noise, and imbalance. Traditional time series classification techniques frequently fall short in addressing these issues, leading to reduced generalization performance. Therefore, there is a need for innovative methodologies to enhance data diversity and quality. In this paper, we introduce a method for the extraction of features for time series classification using noise injection to address these challenges. By employing noise injection techniques for data augmentation, we enhance the diversity of the training data. Utilizing digital signal processing (DSP), we extract key frequency features from time series data through sampling, quantization, and Fourier transformation. This process enhances the quality of the training data, thereby maximizing the model’s generalization performance. We demonstrate the superiority of our proposed method by comparing it with existing time series classification models. Additionally, we validate the effectiveness of our approach through various experimental results, confirming that data augmentation and DSP techniques are potent tools in time series data classification. Ultimately, this research presents a robust methodology for time series data analysis and classification, with potential applications across a broad spectrum of data analysis problems.

## 1. Introduction

Time series data play an important role in various fields such as finance [1], healthcare [2], and industrial processes [3], and accurate analysis of such data is essential for decision-making and performance improvement in these areas. However, due to the complexity and variability of time series data, effectively classifying them remains a challenging task. Time series data are characterized by nonlinearity and non-stationarity over time, and failure to properly capture these characteristics can lead to degraded classification performance. Existing time series classification methods include traditional statistical techniques [4], machine learning algorithms [5], and deep learning-based models [6], but they have several limitations due to data imbalance, noise, and limited data quantities [7]. For example, traditional methods struggle to capture complex patterns in data, and deep learning models require large amounts of labeled data and high computational resources. Therefore, there is a need to develop new classification methodologies that can effectively handle the complexity of time series data and apply them to real-world problems across various domains. In this paper, we propose a novel approach that combines noise injection and digital signal processing techniques to overcome these limitations. Through this method, we enhance the features of the data and demonstrate that high classification performance can be achieved even with small amounts of data.

To address these issues, data augmentation techniques and DSP methods have emerged as useful tools [8]. Data augmentation techniques enhance data diversity by applying various transformations to existing data, thereby improving the generalization performance of models. Notably, noise injection is a technique that generates new data by adding random noise to the original data, significantly increasing data diversity. This allows the model to learn from a wider range of scenarios, thereby improving classification performance [9]. DSP is a method that transforms the features of time series data, enabling more effective extraction and representation of information. Through DSP techniques such as frequency analysis, filtering, and transformation, important features of time series data can be highlighted, or noise can be removed. This enhancement enables classification models to perform better [10].

Therefore, this paper aims to enhance the classification performance of time series data by combining data augmentation through noise injection and DSP methods. This approach enables the model to learn from the diverse variability of time series data and improves classification performance by emphasizing important features. The proposed methodology consists of three main steps. First, data augmentation is performed through noise injection. Second, DSP techniques are applied to the augmented data to extract important features from the time series data. Finally, the transformed features are classified using existing time series data models. This process aims to effectively model the complex patterns and dynamic changes of time series data, achieving high classification accuracy. By providing a detailed methodology, experimental results, and insights gained from this novel approach, this paper comprehensively evaluates the impact of integrating deep learning and DSP techniques on the field of time series data analysis.

The contributions of the proposed method in this paper are as follows:Novel Integration of Noise Injection and DSP for Time Series Classification: We propose a unique methodology that synergistically combines noise injection and digital signal processing techniques specifically for time series classification. This integrated approach is novel, as previous works have explored these techniques individually but not in conjunction.Enhanced Feature Extraction Framework: By applying DSP techniques—including sampling, quantization, and Fourier transformation—we develop an effective framework for extracting salient frequency features from time series data. This improves feature quality and contributes to better model learning and classification performance.Improved Classification Performance with Limited Data: Our method demonstrates that high classification accuracy can be achieved even with limited datasets by increasing data diversity through controlled noise injection. This addresses a common limitation in time series analysis where data collection is challenging.Validation Across Diverse Domains: We validate the generalizability of our approach through experiments on various datasets from the UCR Time Series Classification Archive, covering multiple domains such as biology, healthcare, finance, and industry. This confirms that our method is not limited to a specific dataset or field.

The structure of this paper is as follows. Section 2 describes time series data augmentation and DSP. Section 3 details extraction of features for time series classification using noise injection. Section 4 presents the results and performance evaluation, and Section 5 concludes the paper.

## 2. Related Work

### 2.1. Time Series Data Augmentation

Time series data play a crucial role in various fields, but it is often challenging to secure enough data. This can lead to overfitting due to the lack of data necessary for model training, ultimately reducing the model’s generalization performance. Data augmentation is a critical technique to address this issue [11]. By applying various transformations to existing data, data augmentation artificially increases the size and diversity of the dataset. This enables the model to learn from a wider range of patterns, preventing overfitting and enhancing generalization performance.

Various data augmentation techniques can be applied to time series data, each generating new data by transforming specific aspects of the original data. Shifting involves moving the time series data by a certain time interval, creating new data by altering the temporal position of the series [12]. This makes the model more robust against temporal variations. Scaling entails enlarging or reducing the values of the time series data by a fixed ratio [13], allowing the model to learn from variations in data magnitude. Cropping involves cutting out a portion of the time series data to use as new data [14], increasing the model’s sensitivity to specific parts of the data. Flipping reverses the time series data from left to right, which can be useful for data with symmetrical patterns [15].

Noise injection is a data augmentation technique that generates new data by adding random noise to the original time series data. Noise injection helps models learn robustly despite noise in the data, ensuring stable operation even with noisy data. Magklaras et al. [16] demonstrated that noise injection can protect the integrity of time series data and enable the detection of malicious data by accessing data validity using statistical analysis and machine learning algorithms at the receiving end. Thus, data augmentation techniques, particularly noise injection, play a crucial role in improving the classification performance of time series data. This paper aims to enhance the classification performance of time series data by leveraging these techniques. Figure 1 shows the application of five data augmentation methods, namely, shifting, scaling, cropping, flipping, and noise injection; (a) shows the original data, (b) displays the result of shifting, (c) depicts scaling, (d) demonstrates cropping, (e) presents flipping, and (f) exhibits the result of noise injection.

### 2.2. Digital Signal Processing

DSP is a technology that uses digital computers or electronic devices to analyze and transform signals. DSP can process signals in either the time domain or the frequency domain, playing a crucial role in various applications such as noise removal, feature extraction, and signal compression. DSP is highly useful for analyzing the complex patterns of time series data and extracting important information to enhance data quality. This is particularly beneficial in the classification of time series data, where DSP can significantly improve model performance [17].

DSP encompasses various techniques, each tailored to process signals for specific purposes. Frequency analysis converts time-domain signals to the frequency domain to analyze their frequency components, with the Fourier transform being a representative method [18]. Filtering involves removing or emphasizing specific frequency components of a signal. Common types include low-pass filters, high-pass filters, and band-pass filters [19]. Wavelet transformation is a method that analyzes signals in the time-frequency domain, allowing simultaneous examination of temporal changes and frequency components, making it particularly useful for non-stationary signals [20]. This technique can transform time series data to create new representations, highlighting or modifying specific signal patterns. In this paper, features are extracted through a sequence of processes including sampling, quantization, and Fourier transformation [21,22].

DSP plays a critical role in the analysis and classification of time series data. For example, Ghaderpour et al. [23] reviewed various frequency and time-frequency decomposition methods, enabling the analysis of non-stationary time series and irregularly spaced data, and presented advanced techniques applicable across various scientific fields. Such reviews highlight the significance of DSP as a crucial tool in time series data analysis and classification, enhancing data quality and improving model performance. This paper aims to further enhance the classification performance of time series data by leveraging these DSP techniques.

## 3. Methodology

This paper is composed of three stages. The first stage involves data augmentation through noise injection. The second stage processes the augmented data using DSP. The third stage classifies the data based on the extracted features. Figure 2 shows the overall process of the proposed methodology in this paper.

The first step involves data augmentation through noise injection, where Gaussian noise with a level set at 30% of the standard deviation is used to augment the data tenfold. The next step is feature extraction via DSP, which effectively removes any additional noise introduced during data augmentation. Finally, time series classification is conducted using an existing time series classification model. The classification model employed is LSTM, and its performance is compared and analyzed against baseline models such as the Gated Recurrent Unit (GRU), Temporal Convolutional Network (TCN), and Transformer to demonstrate the superiority of the proposed methodology [24,25,26,27]. Additionally, the study compares the performance of different application orders of DSP and noise injection to optimize the proposed method, evaluates the impact of hyperparameter changes in the DSP stage by varying the intensity of sampling, and conducts ablation studies to analyze various influences. This comprehensive evaluation highlights the distinctiveness of the proposed methodology.

### 3.1. Data Augmentation Using Noise Injection

Data augmentation is a crucial technique for enhancing the generalization performance of models, especially when there is a lack of sufficient data. In this paper, we improve classification performance by increasing the diversity of time series data through noise injection. Noise injection involves adding random noise to the original data to generate new data, allowing the model to learn various variations and improve generalization performance. Noise injection is a data augmentation technique that generates new data by adding a specific level of random noise to the original data. In this paper, Gaussian noise was used, and the noise level was set to 30% of the standard deviation [28]. Due to the varying data distributions across datasets, using a fixed noise level is less effective. Instead, setting the noise level based on the standard deviation of each dataset allows for more effective data augmentation. This involves adding random values, generated based on the mean and standard deviation of the original data, to each data point. The following formula represents this process:(1)Noise Level=0.3×σ
where σ represents the standard deviation of the data.

In this paper, we generate tenfold-augmented data for each original data point. This means that for each sample in the original dataset, nine additional samples are created, effectively increasing the size of the entire dataset by a factor of ten. This augmentation ratio allows the model to learn from a variety of patterns and variations, thereby preventing overfitting and enhancing generalization performance. Figure 3 shows the crop data from the UCRArchive2018 dataset before and after augmentation [29].

The results in Figure 3 show that the augmented data retain a similar shape to the original data, with slight perturbations. This confirms the effectiveness of the noise injection technique in generating realistic yet diverse data samples. These visual comparisons highlight that while the augmented data are like the original, the noise injection introduces subtle variations, contributing to increased data diversity and enhanced model generalization performance.

### 3.2. Time Series Feature Extraction Using Signal Processing

DSP is a critical technology used to enhance data quality and extract important information through signal analysis, transformation, and feature extraction. In this paper, DSP is applied in three stages—sampling, quantization, and discrete Fourier transformation (DFT)—to improve the classification performance of time-series data. Each stage addresses specific aspects of signal processing and ultimately contributes to the extraction of high-quality features, maximizing the model’s performance.

#### 3.2.1. Sampling

Sampling is the first stage of converting an analog signal into a digital signal, which involves measuring continuous-time domain signals at regular intervals to transform them into discrete signals. In this paper, experiments confirmed that a sampling rate of 0.5 is most suitable, and it was set as the final sampling rate [30]. A sampling rate of 0.5 captures the frequency components of the signal sufficiently while managing the data volume efficiently. The appropriate selection of the sampling rate maintains the information of the data without loss while preventing the excessive generation of unnecessary data. Figure 4 shows the application of the sampling process to the crop data.

Figure 4 shows (a) the augmented data and (b) the augmented data after applying the sampling process. While there are no significant changes in the data, it can be observed that the overall curvature becomes smoother.

#### 3.2.2. Quantization

Quantization is the process of converting the amplitude values of sampled discrete signals into digital values by dividing them into fixed levels. In this paper, 16-bit quantization is used to convert the signals into digital values. The 16-bit quantization uses 65,536 (2^16^) distinct values to represent the signal amplitude, which allows for sufficient detail representation of the signal while efficiently managing data size [31]. The quantization process is designed to minimize quantization noise. Figure 5 shows the application of the quantization process to the crop data after the sampling process.

Figure 5 shows (a) the data after applying the sampling process and (b) the data after applying both sampling and quantization processes. Since the quantization value is set at 16 bits, which is a very high quantization level, the actual data values do not change significantly. In the case of the crop data, the original values change very slightly, such as from 0.240 to 0.240003 and from 0.219 to 0.219003.

#### 3.2.3. Discrete Fourier Transform

Fourier transformation is a method for converting a time-domain signal into the frequency domain to analyze the signal’s frequency components. In this paper, the DFT is used to extract the frequency features of the sampled and quantized time-series data [32]. The DFT is calculated using the following equation:(2)$$X\left(k\right)=\sum_{n=0}^{N-1}x(n)\times e^{-j2\pi kn/N}$$
where X(k) represents the DFT of the signal, x(n) is the sampled and quantized signal, N is the number of points in the signal, and k is the index of the frequency component. This transformation allows for the identification of significant frequency components within the data. The frequency components obtained through DFT contain important frequency features of the time-series data, which are used for model training. Figure 6 shows the results after applying all DSP steps to the crop data.

Figure 6 shows (a) the original crop data and (b) the final processed data after applying all DSP steps. It can be observed that most of the feature values have changed significantly from the original data. Algorithm 1 shows the DSP process used in this paper. It consists of sampling, quantization, and DFT.
**Algorithm 1** Digital Signal Processing**Require**: signal (signal data)**Ensure**: magnitude, phase1: **function** DFT(signal)2:   dft_result <- FFT(signal)3:   frequencies <- FFT_freq(signal)4:   magnitude <- abs(dft_result)5:   phase <- angle(dft_result)6:   **return** (frequencies, magnitude, phase)7: **end function**
8: **function** QUANTIZE(data, levels)9:   min_val <- min(data)10:  max_val <- max(data)11:  intervals <- linspace(min_val, max_val, levels-1)12:  quantized_data <- digitize(data, intervals)13:  mapped_values <- (max_val-min_val)/(levels-1)*quantized_data+min_val14:  **return** mapped_values15: **end function**
16: **function** DFT_TO_FEATURES(signal)17:  (magnitude, phase) <- DFT(signal)18:  **return** (magnitude, phase)19: **end function**
20: **procedure** DSP(signal)21:  sampling_rate <- 0.522:  sample_count <- length(signal)*sampling_rate23:  sampled_indices <- linspace(0, length(signal)-1,sample_count)
24:  sampled_data <- signal[sampled_indices]25:  quantization_levels <- $2^{16}$
26:  quantized_data <- QUANTIZE(sampled_data, quantization_levels)27:  (magnitude, phase) <- DFT_TO_FEATURES(sampled_data)28:  **return** (magnitude, phase)29: **end procedure**

### 3.3. Time Series Classification Based on LSTM

Time series classification (TSC) is a critical problem in various real-world applications [33]. In this paper, a vanilla long short-term memory (LSTM) model is used to perform time series classification [34]. LSTM, a type of recurrent neural network (RNN), is particularly strong in learning dependencies in long sequence data [35]. In this paper, the LSTM model is employed to classify time series data, and techniques such as data augmentation and DSP are applied to enhance the model’s performance.

In this paper, a vanilla LSTM model is used to perform time series classification. LSTM, a type of RNN, can learn dependencies in long sequence data. The key feature of LSTM is its cell state and gate structure, which allows it to selectively remember and forget information. This capability enables LSTM to effectively learn from long sequence data.

The model training is conducted for 300 epochs without pretraining, applying a tenfold augmentation and DSP to the training data. In contrast, only DSP is applied to the test data to evaluate the model’s performance. The Adam optimizer is used for each epoch to minimize the model’s loss function. Figure 7 shows the structure of the LSTM used in this paper.

## 4. Result and Performance Evaluation

### 4.1. Dataset and Performance Evaluation Metrices for Experiments

In this section, the dataset and data augmentation techniques are described. All experiments were conducted on a system equipped with an Intel^®^ Xeon^®^ Silver 2.40 GHz CPU, 128 GB RAM, and a single NVIDIA RTX 3090 GPU.

The performance evaluation in this paper was conducted using the UCRArchive2018 dataset. The UCR is a comprehensive collection of time series datasets used for analysis and classification, encompassing various domains. This dataset is widely employed for comparing and validating the performance of classification algorithms. One of the key features of the UCR dataset is that it includes dozens of datasets, each reflecting the characteristics of different applications and data types. This diversity is highly beneficial for evaluating the generalization ability of algorithms. However, due to the vast number of datasets, making efficient selections for experimentation is essential. In this paper, the following seven major categories are used to effectively categorize and select datasets from the UCR repository: image outline, sensor readings, motion capture, spectrographs, electric devices, ECG, and simulated. By selecting datasets that fit each of these categories, the experimental data can be utilized more efficiently for the intended purpose of the study. Categorizing the diverse UCR datasets in this manner allows for the evaluation of how consistently each algorithm performs across time series data with different characteristics. The datasets chosen from each category are split into training and testing sets, with 20% of the training data used for validation during the training process. All experiments were conducted using untrained models with the number of epochs fixed at 300.

The evaluation metric used in the experiments was accuracy. Accuracy indicates how accurately the model predicts the class of the input image and is calculated as the proportion of correctly classified data out of the total data [36]. The accuracy is computed using the following formula:(3)$$Accuracy=\frac{TP+TN}{TP+FP+TN+FN}$$

Accuracy indicates how accurately the model predicts the class of the input image and is calculated as the proportion of correctly classified data out of the total data. Equation (3) shows the method for calculating accuracy [37,38]. TP (true positive) refers to cases where the model correctly predicts a positive instance as positive. TN (true negative) refers to cases where the model correctly predicts a negative instance as negative. FP (false positive) refers to cases where the model incorrectly predicts a negative instance as positive. FN (false negative) refers to cases where the model incorrectly predicts a positive instance as negative.

### 4.2. Proposed Methods Performance Evaluation

In Section 4.2, we present a comparative performance analysis between the proposed method and existing models. The evaluation metric used for the comparison is accuracy, which represents the proportion of correctly predicted instances out of the total instances. The accuracy results of the experiments are as follows. Table 1 shows the performance evaluation results of the proposed method.

Table 1’s “Normal” represents the performance evaluation of LSTM on each UCR dataset without any preprocessing. The results in Table 1 indicate that our methodology achieves higher performance across various datasets. This demonstrates that noise injection and DSP are effective for time series data classification. Specifically, analysis of the datasets with the most significant performance improvements reveals that our method shows substantial gains in few-shot data and binary classification problems. This finding substantiates the effectiveness of the proposed methodology for few-shot data and binary classification tasks. Conversely, the analysis of datasets where performance did not improve or even declined shows that these typically involve a high number of features and multi-class classification problems. For instance, the CinCECGTorso dataset has 1640 features and four classes. Future work will aim to develop methodologies that are effective for datasets with many features and multi-class classification problems to address these limitations.

### 4.3. Performance Comparison with Variations in DSP Hyperparameters

In this section, we evaluate the impact of changes in the hyperparameters of the DSP stage on performance by comparing performance across different sampling intensities. This comparison focuses on exploring the flexibility of DSP and its applicability across various scenarios. The experiments were conducted by varying the sampling intensity from 0.1 to 0.9. The datasets with the most significant performance improvements—crop, FordA, and trace—were used for the experiments. Table 2 shows the performance comparison based on changes in the DSP hyperparameters.

The experimental results indicate that the highest performance is achieved when the sampling intensity is set between 0.4 and 0.6. When the sampling intensity is too low or too high, it can result in reduced representativeness of the data or overfitting due to excessive sampling. A sampling intensity between 0.4 and 0.6 provides adequate data diversity, enhancing the model’s generalization capability. Within this range, data diversity is sufficiently maintained, while important features are preserved, allowing the model to learn effectively. This finding can serve as a crucial criterion for optimizing sampling intensity in future work.

### 4.4. Comparison of Application Order Performance of Noise Injection and DSP

In this section, we evaluate the impact of the application order of DSP and noise injection on performance by comparing the performance based on different sequences. The experiments were conducted under two conditions. The first condition involved applying DSP before performing noise injection augmentation, and the second condition involved performing noise injection augmentation before applying DSP. Table 3 shows the performance comparison based on the application order.

Applying DSP after augmentation optimizes model performance by first increasing data diversity through data augmentation and then enhancing the signal and removing noise using DSP techniques. This sequence allows DSP to effectively eliminate additional noise introduced during data augmentation, resulting in more refined data for training. Conversely, applying data augmentation after DSP can reintroduce noise that was removed in the DSP stage, potentially degrading the quality of the final training data and negatively impacting model performance. Consequently, it is confirmed that applying DSP after data augmentation achieves higher performance. This finding suggests the optimal application sequence of noise injection and DSP techniques for time series data, providing a basis for further performance improvements in future work.

### 4.5. Comparison with Time Series Models

For Section 4.5, we conducted comparative experiments with various state-of-the-art time series models to evaluate the performance of the proposed methodology. The models selected for comparison included LSTM, GRU, TCN, and Transformer. By assessing the performance of each model, we demonstrated the superiority of the proposed methodology. Table 4 shows the comparative performance of different time series models.

The results in Table 4 show that the proposed methodology outperforms traditional models such as LSTM, GRU, TCN, and Transformer. The superior performance of the proposed methodology highlights the effectiveness of noise injection augmentation and DSP application for time series data classification. This finding suggests a promising approach for future work in time series analysis.

### 4.6. Ablation Study

For this section, we applied noise injection augmentation and DSP techniques to enhance the performance of time series data classification. To analyze the impact of each technique on model performance, we conducted an ablation study. The experiments were carried out in the following order: vanilla model, DSP application, augmentation application, and the application of both DSP and augmentation. This approach allowed us to evaluate the independent and combined effects of each technique. Table 5 shows the results of the ablation study.

The experimental results indicate that both the application of DSP and augmentation led to performance improvements compared to the vanilla model. Notably, the highest performance improvement was observed when both techniques were applied together. The application of DSP enhances model performance by removing noise from time series data and emphasizing key features of the signal, resulting in an average performance improvement of approximately 9.33% compared to the vanilla model. Additionally, data augmentation increases the diversity of the training data, thereby improving the model’s generalization ability, with an average performance improvement of approximately 8.37% compared to the vanilla model.

When both DSP and data augmentation techniques are applied, the combined advantages of both methods yield the highest performance improvement. This outcome demonstrates that enhancing the signal during the preprocessing stage and simultaneously increasing the diversity of the training data can maximize the learning effectiveness of the model. Specifically, the combined application of both techniques results in an average performance improvement of approximately 16.81% compared to the vanilla model. These findings support the validity of the proposed methodology and demonstrate that DSP and data augmentation techniques play a crucial role in enhancing the performance of time series data classification.

## 5. Conclusions

In this paper, we proposed a new methodology that combines data augmentation and digital signal processing techniques to enhance the performance of time series data classification. For data augmentation, we employed a noise injection technique to increase the diversity of the training data, augmenting each data sample tenfold by setting the Gaussian noise level to 30% of the standard deviation. Additionally, DSP was conducted through three stages—sampling, quantization, and Fourier transformation—to extract important frequency features of the time series data. This approach improved the quality of the training data and maximized the model’s generalization performance.

The experimental results showed that the proposed methodology significantly improved time series data classification performance compared to existing methods, demonstrating superior performance across various evaluation metrics. These findings confirm that the combination of data augmentation and DSP techniques is an effective tool for addressing time series data classification problems.

The results of this paper have the following implications. First, the integration of data augmentation and DSP techniques supports the effective learning of complex patterns in time series data, indicating that it can overcome limitations caused by data imbalances and noise. Second, achieving high classification performance with a small amount of data suggests potential applicability in fields where data collection is challenging. Furthermore, this paper contributes to existing research as follows. While existing time series classification methods have mainly focused on individual techniques, our paper confirmed the synergistic effect of combining data augmentation and DSP. This presents a new approach in the field of time series data classification and contributes to indicating the direction for future research.

In future work, we plan to further explore the scalability and applicability of the proposed methodology. We aim to develop effective methods for problems with many features and multi-class classification, thereby constructing a generalized model applicable to various time series datasets. Additionally, we will proceed with research to further enhance model performance by combining more complex signal processing techniques and advanced data augmentation methods. Through this, we expect to make practical contributions to solving time series data classification problems.

## Figures and Tables

**Figure 1 sensors-24-06402-f001:**
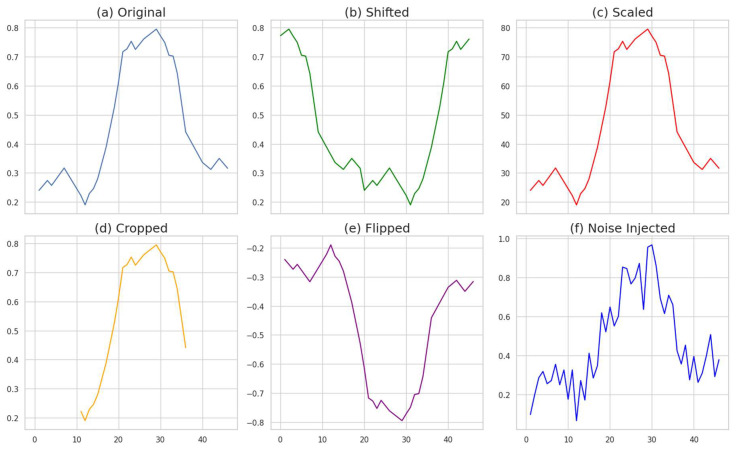
Five data augmentation results.

**Figure 2 sensors-24-06402-f002:**
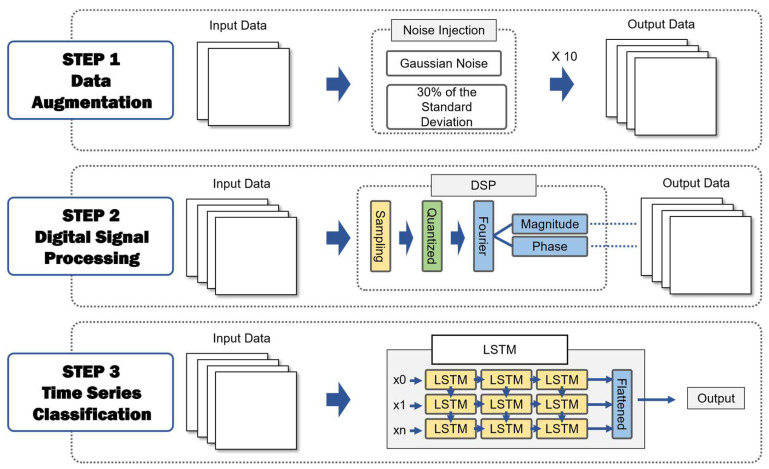
Process of extraction of features for time series classification using noise injection.

**Figure 3 sensors-24-06402-f003:**
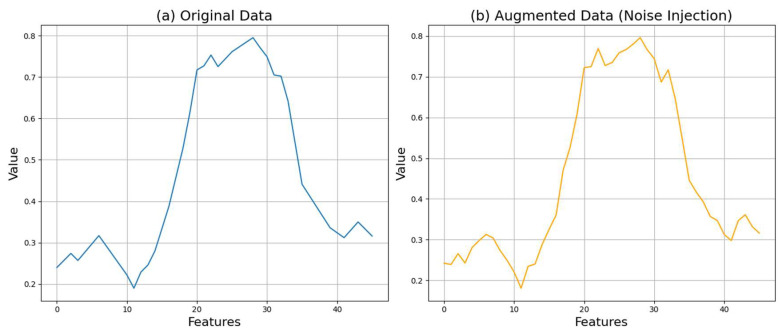
Comparison of crop data before and after noise injection.

**Figure 4 sensors-24-06402-f004:**
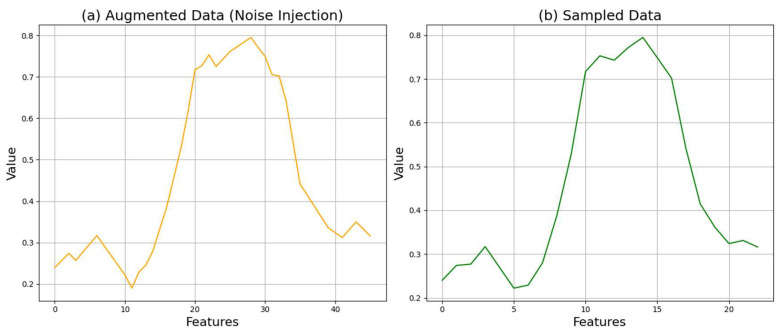
Comparison of crop data before and after sampling.

**Figure 5 sensors-24-06402-f005:**
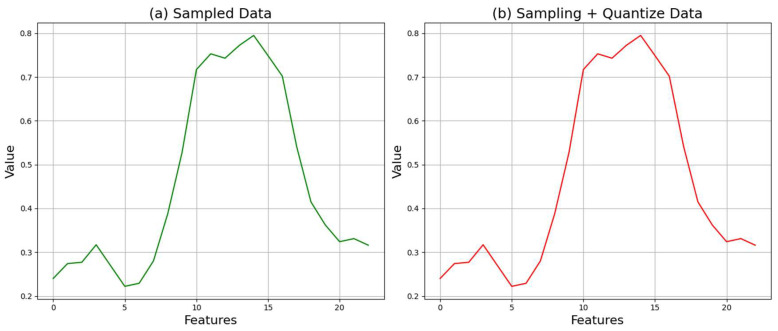
Comparison of crop data before and after quantization.

**Figure 6 sensors-24-06402-f006:**
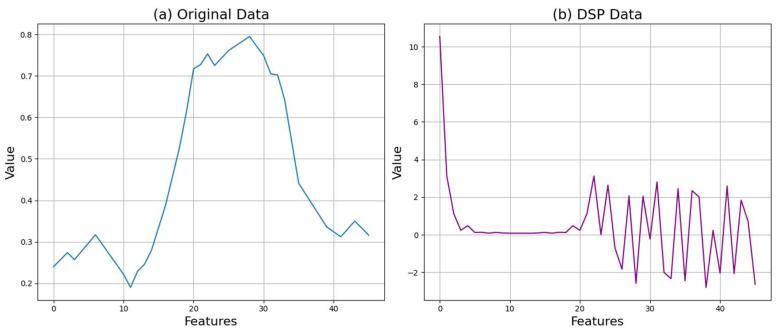
Comparison of crop data before and after discrete Fourier transformation.

**Figure 7 sensors-24-06402-f007:**
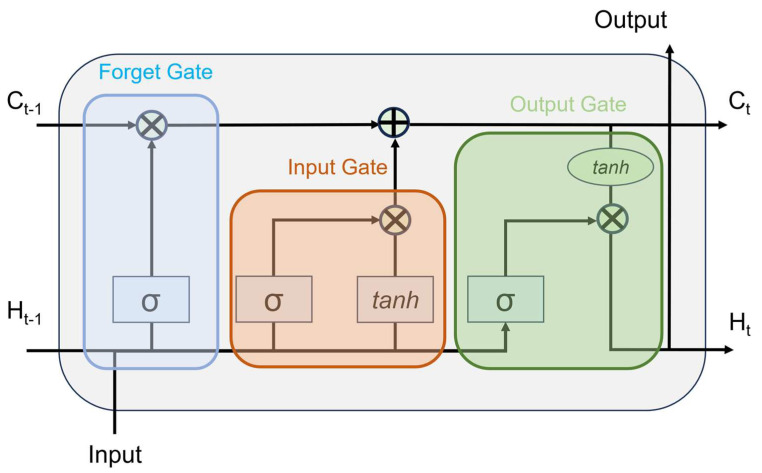
Structure of the LSTM.

**Table 1 sensors-24-06402-t001:** Performance evaluation of proposed methods.

Category	Data	LSTM	
		Normal	Ours
Image Outline	Crop	63.26%	72.80%
	ShapesAll	69.50%	72.17%
	InlineSkate	30.55%	33.09%
Sensor readings	DodgerLoopGame	52.17%	52.17%
	DodgerLoopWeekend	73.91%	73.91%
	FordA	78.33%	94.24%
Motion capture	Wafer	99.03%	99.48%
	CricketX	44.87%	54.87%
Spectrographs	Trace	63.0%	88.0%
	ChlorineConcentration	58.80%	73.59%
Electric devices	SonyAIBORobotSurface1	80.03%	83.86%
	SonyAIBORobotSurface2	82.69%	85.67%

**Table 2 sensors-24-06402-t002:** Performance comparison with variations in DSP hyperparameters.

	Crop	FordA	Trace
0.1	50.51%	82.80%	81.00%
0.2	66.71%	92.27%	87.00%
0.3	67.68%	93.79%	85.00%
0.4	69.12%	94.17%	88.00%
0.5	72.80%	94.24%	85.00%
0.6	72.42%	93.48%	83.00%
0.7	72.30%	93.79%	82.00%
0.8	71.74%	94.02%	83.00%
0.9	72.79%	93.94%	82.00%

**Table 3 sensors-24-06402-t003:** Comparison of application order performance of noise injection and DSP.

	Crop	FordA	Trace
DSP → Aug	68.76%	93.79%	76.00%
Aug → DSP	72.80%	94.24%	88.00%

**Table 4 sensors-24-06402-t004:** Comparison between time series models.

	Crop	FordA	Trace
LSTM [24]	63.26%	78.33%	63.00%
GRU [25]	61.29%	79.92%	43.00%
TCN [26]	43.54%	79.39%	68.00%
Transformer [27]	68.69%	74.62%	68.00%
Ours	72.80%	94.24%	88.00%

**Table 5 sensors-24-06402-t005:** Ablation study.

	LSTM	DSP	Aug	DSP + Aug
Crop	63.26%	66.70%	71.95%	72.80%
FordA	78.33%	92.99%	79.77%	94.24%
Trace	63.00%	73.00%	78.00%	88.00%

## Data Availability

The data presented in this study are available in the article.

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
