# Peer review of "Extraction of Features for Time Series Classification Using Noise Injection"

_sensors, 2024, doi:10.3390/s24196402_

Round 1
Reviewer 1 Report
Comments and Suggestions for Authors
In this paper, the authors introduced a method for extracting features for time series classification using the noise injection method to address these challenges. While the paper is well written with sufficient data and results provided, it is crucial to make the following aspects more straightforward for the readers' understanding:
1. Why is only a noise level of 0.01 used? A reference work proposes this level, so what is the main difference between the work and this paper?
2. It is highly recommended that the authors consider experimenting with different noise levels. This exploration could lead to the discovery of the most optimized noise level for this approach, potentially enhancing the method's effectiveness and broadening its applicability.
3. It is crucial to provide more information about the datasets used in Section 4.2. This will not only enhance the transparency of your research but also help readers better understand and replicate your experiments. For instance, where are the Normal values (in Table 1) collected?
4. It is important to give references for data in Table 4 (results of LSTM, GRU, TCN, Transformer). If these values are extracted from your execution without proper references, the comparisons will seem unfair. Rectifying this will ensure the credibility of your research.
5. Since the quantization is applied, the performance of the proposed approach (in terms of execution time) should be improved. Please provide more data about performance compared to the others in the literature.
Comments on the Quality of English LanguageGood enough
Author Response
Comments 1: Why is only a noise level of 0.01 used? A reference work proposes this level, so what is the main difference between the work and this paper?
Response 1: Thank you for your comment. The UCR dataset that We are using contains a variety of data types with different distributions. Due to these variations, instead of using a fixed noise level of 0.01, We set the noise level to 30% of the standard deviation of each dataset. The details of this modification are outlined below.
The first step involves data augmentation through noise injection, where Gaussian noise with a level set at 30% of the standard deviation is used to augment the data tenfold. The next step is featuring extraction via DSP, which effectively removes any additional noise introduced during data augmentation. Finally, Time Series Classification is conducted using an existing time series classification model. The classification model employed is LSTM, and its performance is compared and analyzed against baseline models such as Gated Recurrent Unit (GRU), Temporal Convolutional Network (TCN), and Transformer to demonstrate the superiority of the proposed methodology [20-23]. Additionally, the study compares the performance of different application orders of DSP and noise injection to optimize the proposed method, evaluates the impact of hyperparameter changes in the DSP stage by varying the intensity of sampling, and conducts ablation studies to analyze various influences. This comprehensive evaluation highlights the distinctiveness of the proposed methodology.
3.1. Data Augmentation using Noise Injection
Data augmentation is a crucial technique for enhancing the generalization performance of models, especially when there is a lack of sufficient data. In this paper, we improve classification performance by increasing the diversity of time series data through noise injection. Noise injection involves adding random noise to the original data to generate new data, allowing the model to learn various variations and improve generalization performance.
Noise injection is a data augmentation technique that generates new data by adding a specific level of random noise to the original data. In this paper, Gaussian noise was used, and the noise level was set to 30% of the standard deviation [24]. Due to the varying data distributions across datasets, using a fixed noise level is less effective. Instead, setting the noise level based on the standard deviation of each dataset allows for more effective data augmentation. This involves adding random values, generated based on the mean and standard deviation of the original data, to each data point. The following formula represents this process:
|
|
(1) |
where represents the standard deviation of the data.
- Conclusions
In this paper, we proposed a novel methodology that combines data augmentation and DSP techniques to improve the performance of time series data classification. For data augmentation, we employed a noise injection technique to enhance the diversity of the training data, setting the Gaussian noise level to 30% of the standard deviation and augmenting each data sample by a factor of 10. Additionally, the DSP process was carried out in three stages: sampling, quantization, and Fourier transformation, to extract significant frequency features from the time series data. This approach enhanced the quality of the training data and maximized the generalization performance of the model.
Comments 2: It is highly recommended that the authors consider experimenting with different noise levels. This exploration could lead to the discovery of the most optimized noise level for this approach, potentially enhancing the method's effectiveness and broadening its applicability.
Response 2: Thank you for your comment. The noise level is a hyperparameter used for data augmentation, and unless it is excessively large or small, it does not significantly impact performance. Therefore, We set the noise level to 30% of the standard deviation of each dataset, and even when adjusted to 20% or 40%, no significant performance differences were observed.
Comments 3: It is crucial to provide more information about the datasets used in Section 4.2. This will not only enhance the transparency of your research but also help readers better understand and replicate your experiments. For instance, where are the Normal values (in Table 1) collected?
Response 3: Thank you for your comment. The following corrections have been made.
4.1. Dataset and Performance Evaluation Metrices for Experiments
In this section, the dataset and data augmentation techniques are described. All experiments are conducted on a system equipped with an Intel® Xeon® Silver 2.40 GHz CPU, 128GB RAM, and a single NVIDIA RTX 3090 GPU.
The performance evaluation in this paper is conducted using the UCRArchive2018 dataset. The UCR is a comprehensive collection of time series datasets used for analysis and classification, encompassing various domains. This dataset is widely employed for comparing and validating the performance of classification algorithms. One of the key features of the UCR dataset is that it includes dozens of datasets, each reflecting the characteristics of different applications and data types. This diversity is highly beneficial for evaluating the generalization ability of algorithms. However, due to the vast number of datasets, making efficient selections for experimentation is essential. In this paper, the following seven major categories are used to effectively categorize and select datasets from the UCR repository: Image Outline, Sensor Readings, Motion Capture, Spectrographs, Electric Devices, ECG, and Simulated. By selecting datasets that fit each of these categories, the experimental data can be utilized more efficiently for the intended purpose of the study. Categorizing the diverse UCR datasets in this manner allows for the evaluation of how consistently each algorithm performs across time series data with different characteristics. The datasets chosen from each category are split into training and testing sets, with 20% of the training data used for validation during the training process. All experiments are conducted using untrained models with the number of epochs fixed at 300.
Table 1. Proposed Methods Performance Evaluation.
|
Category |
Data |
LSTM |
|
|
Normal |
Ours |
||
|
Image Outline |
Crop |
63.26% |
72.80% |
|
ShapesAll |
69.50% |
72.17% |
|
|
InlineSkate |
30.55% |
33.09% |
|
|
Sensor readings |
DodgerLoopGame |
52.17% |
52.17% |
|
DodgerLoopWeekend |
73.91% |
73.91% |
|
|
FordA |
78.33% |
94.24% |
|
|
Motion capture |
Wafer |
99.03% |
99.48% |
|
CricketX |
44.87% |
54.87% |
|
|
Spectrographs |
Trace |
63.0% |
88.0% |
|
ChlorineConcentration |
58.80% |
73.59% |
|
|
Electric devices |
SonyAIBORobotSurface1 |
80.03% |
83.86% |
|
SonyAIBORobotSurface2 |
82.69% |
85.67% |
|
Table 1's "Normal" represents the performance evaluation of LSTM on each UCR dataset without any preprocessing. The results in Table 1. indicate that our methodology achieves higher performance across various datasets. This demonstrates that noise injection and DSP are effective for time series data classification. Specifically, analysis of the datasets with the most significant performance improvements reveals that our method shows substantial gains in few-shot data and binary classification problems. This finding substantiates the effectiveness of the proposed methodology for few-shot data and binary classification tasks. Conversely, the analysis of datasets where performance did not improve or even decline shows that these typically involve a high number of features and multi-class classification problems. For instance, the CinCECGTorso dataset has 1640 features and 4 classes. Future work will aim to develop methodologies that are effective for datasets with many features and multi-class classification problems to address these limitations.
Comments 4: It is important to give references for data in Table 4 (results of LSTM, GRU, TCN, Transformer). If these values are extracted from your execution without proper references, the comparisons will seem unfair. Rectifying this will ensure the credibility of your research.
Response 4: Thank you for your comment. The following corrections have been made.
Table 4. Comparison with Time Series Models.
|
|
Crop |
FordA |
Trace |
|
LSTM [20] |
63.26% |
78.33% |
63.00% |
|
GRU [21] |
61.29% |
79.92% |
43.00% |
|
TCN [22] |
43.54% |
79.39% |
68.00% |
|
Transformer [23] |
68.69% |
74.62% |
68.00% |
|
Ours |
72.80% |
94.24% |
88.00% |
- Karim, F.; Majumdar, S.; Darabi, H.; Chen, S. LSTM fully convolutional networks for time series classification. IEEE access. 2017, 6, 1662-1669.
- Elsayed, N.; Maida, A.S.; Bayoumi, M. Gated recurrent neural networks empirical utilization for time series classification. In Proceedings of the International Conference on Internet of Things (iThings) and IEEE Green Computing and Communications (GreenCom) and IEEE Cyber, Physical and Social Computing (CPSCom) and IEEE Smart Data (SmartData), Atlanta, GA, USA, 14-17 July 2019; pp. 1207-1210.
- Koh, B.H.D.; Lim, C.L.P.; Rahimi, H.; Woo, W.L.; Gao, B. Deep temporal convolution network for time series classification, 2021, 21, 603.
- Zerveas, G.; Jayaraman, S.; Patel, D.; Bhamidipaty, A.; Eickhoff, C. A transformer-based framework for multivariate time series representation learning. In Proceedings of the 27th ACM SIGKDD Conference on Knowledge Discovery & Data Mining, Singapore, 14-18 August 2021; pp. 2114-2124.
Comments 5: Since the quantization is applied, the performance of the proposed approach (in terms of execution time) should be improved. Please provide more data about performance compared to the others in the literature.
Response 5: Thank you for your comment. Although we applied quantization to our method, it was not intended to significantly compress the data size during the DSP process to reduce execution time. Instead, our focus was on improving classification performance. Additionally, since we increased the data volume tenfold through data augmentation, the training and inference times may be somewhat longer compared to the 'Normal' data that underwent no preprocessing. The main contribution of this paper lies in enhancing classification accuracy rather than execution time, and thus we concentrated on classification performance in our evaluations. Thank you for highlighting the importance of providing additional comparative data on execution time; we will consider this in future research.

Reviewer 2 Report
Comments and Suggestions for Authors
The introduction (lines 27-36) mentions the important role of time series data in a number of fields and points out the limitations of traditional time series classification methods. However, the introduction to the background could have been more detailed and included a more extensive literature review to show the gap between current research and existing research. It is recommended that the authors add more citations of relevant literature in the introduction section to support the need and innovation of the study.
The study design (lines 37-56) describes the enhancement of classification performance of time series data through noise injection and digital signal processing (DSP) techniques. The design appears to be appropriate as it proposes a solution to the complexity and variability of time series data.
The methods section (lines 151-171) describes in detail the three main steps: noise injection, DSP processing and LSTM-based time series classification. Each step is described in detail, including the techniques and parameters used. This part is clearly written.
The results section (lines 266-314) clearly presents the experimental results, including classification performance evaluation using different datasets. Tables and graphs help to understand the results, but it is recommended that the authors provide more detailed information about the experimental setup and datasets.
The conclusions section (lines 393-406) summarises the main findings of the study and points out the effectiveness of data augmentation and DSP techniques in classifying time series data. The conclusions appear to be supported by the results, but it is recommended that the authors further discuss the implications of the results and contributions to existing research.
Overall, the English language quality of the article is good and clear. However, further grammar and spelling proofreading may be required in some places to ensure professionalism.
The study presents a new approach that combines noise injection and DSP techniques, which is novel in the field of time series classification. However, it is recommended that the authors more clearly emphasise the uniqueness of their approach compared to existing techniques.
This research is important for the field of time series data classification as it provides a new approach that may improve classification performance. This is potentially valuable for practical applications in areas such as finance, healthcare and industry.
The research methodology and experimental design appear to be scientifically sound, but it is recommended that the authors provide more detailed information on experimental validation and dataset selection to enhance the credibility of the study.
The study presents a new approach that has the potential to improve the classification performance of time series data. The article is clearly structured, the methodology section is detailed, and the results section provides adequate data support. These strengths indicate that the study is a valuable contribution.
Author Response
Comments 1: The introduction (lines 27-36) mentions the important role of time series data in a number of fields and points out the limitations of traditional time series classification methods. However, the introduction to the background could have been more detailed and included a more extensive literature review to show the gap between current research and existing research. It is recommended that the authors add more citations of relevant literature in the introduction section to support the need and innovation of the study.
Response 1: Thank you for your comment. The following corrections have been made.
- Introduction
Time series data play an important role in various fields such as finance [1], healthcare [2], and industrial processes [3], and accurate analysis of such data is essential for decision-making and performance improvement in these areas. However, due to the complexity and variability of time series data, effectively classifying them remains a challenging task. Time series data are characterized by nonlinearity and non-stationarity over time, and failure to properly capture these characteristics can lead to degraded classification performance. Existing time series classification methods include traditional statistical techniques [4], machine learning algorithms [5], and deep learning-based models [6], but they have several limitations due to data imbalance, noise, and limited data quantities [7]. For example, traditional methods struggle to capture complex patterns in the data, and deep learning models require large amounts of labeled data and high computational resources. Therefore, there is a need to develop new classification methodologies that can effectively handle the complexity of time series data and apply them to real-world problems across various domains. In this paper, we propose a novel approach that combines noise injection and digital signal processing techniques to overcome these limitations. Through this method, we enhance the features of the data and demonstrate that high classification performance can be achieved even with small amounts of data.
- Yuan, A.E., Shoi, W.A rigorous and versatile statistical test for correlations between stationary time series. PLoS Biol. 2024, 22, e3002758.
- Masini, R.P., Medeiros, M.C., Mendes, E. F. Machine learning advances for time series forecasting. Econ. Surv. 2023, 37, 76-111.
- Fawa, H.I., Forestier, G., Weber, J., Idoumghar, L., Muller, P.A. Deep learning for time series classification: a review. Data Min Knowl Disco. 2019, 33, 917-963.
- Dhar, M., Dickinson, J.A., Berg, M.A. Efficient, nonparametric removal of noise and recovery of probability distributions from time series using nonlinear-correlation functions: Additive noise. Chem. Phys. 2023, 159, 054110.
Comments 2: The study design (lines 37-56) describes the enhancement of classification performance of time series data through noise injection and digital signal processing (DSP) techniques. The design appears to be appropriate as it proposes a solution to the complexity and variability of time series data.
Response 2: Thank you for your comment.
Comments 3: The methods section (lines 151-171) describes in detail the three main steps: noise injection, DSP processing and LSTM-based time series classification. Each step is described in detail, including the techniques and parameters used. This part is clearly written.
Response 3: Thank you for your comment.
Comments 4: The results section (lines 266-314) clearly presents the experimental results, including classification performance evaluation using different datasets. Tables and graphs help to understand the results, but it is recommended that the authors provide more detailed information about the experimental setup and datasets.
Response 4: Thank you for your comment. The following corrections have been made.
4.1. Dataset and Performance Evaluation Metrices for Experiments
In this section, the dataset and data augmentation techniques are described. All experiments are conducted on a system equipped with an Intel® Xeon® Silver 2.40 GHz CPU, 128GB RAM, and a single NVIDIA RTX 3090 GPU.
The performance evaluation in this paper is conducted using the UCRArchive2018 dataset. The UCR is a comprehensive collection of time series datasets used for analysis and classification, encompassing various domains. This dataset is widely employed for comparing and validating the performance of classification algorithms. One of the key features of the UCR dataset is that it includes dozens of datasets, each reflecting the characteristics of different applications and data types. This diversity is highly beneficial for evaluating the generalization ability of algorithms. However, due to the vast number of datasets, making efficient selections for experimentation is essential. In this paper, the following seven major categories are used to effectively categorize and select datasets from the UCR repository: Image Outline, Sensor Readings, Motion Capture, Spectrographs, Electric Devices, ECG, and Simulated. By selecting datasets that fit each of these categories, the experimental data can be utilized more efficiently for the intended purpose of the study. Categorizing the diverse UCR datasets in this manner allows for the evaluation of how consistently each algorithm performs across time series data with different characteristics. The datasets chosen from each category are split into training and testing sets, with 20% of the training data used for validation during the training process. All experiments are conducted using untrained models with the number of epochs fixed at 300.
Table 1. Proposed Methods Performance Evaluation.
|
Category |
Data |
LSTM |
|
|
Normal |
Ours |
||
|
Image Outline |
Crop |
63.26% |
72.80% |
|
ShapesAll |
69.50% |
72.17% |
|
|
InlineSkate |
30.55% |
33.09% |
|
|
Sensor readings |
DodgerLoopGame |
52.17% |
52.17% |
|
DodgerLoopWeekend |
73.91% |
73.91% |
|
|
FordA |
78.33% |
94.24% |
|
|
Motion capture |
Wafer |
99.03% |
99.48% |
|
CricketX |
44.87% |
54.87% |
|
|
Spectrographs |
Trace |
63.0% |
88.0% |
|
ChlorineConcentration |
58.80% |
73.59% |
|
|
Electric devices |
SonyAIBORobotSurface1 |
80.03% |
83.86% |
|
SonyAIBORobotSurface2 |
82.69% |
85.67% |
|
Table 1's "Normal" represents the performance evaluation of LSTM on each UCR dataset without any preprocessing. The results in Table 1. indicate that our methodology achieves higher performance across various datasets. This demonstrates that noise injection and DSP are effective for time series data classification. Specifically, analysis of the datasets with the most significant performance improvements reveals that our method shows substantial gains in few-shot data and binary classification problems. This finding substantiates the effectiveness of the proposed methodology for few-shot data and binary classification tasks. Conversely, the analysis of datasets where performance did not improve or even decline shows that these typically involve a high number of features and multi-class classification problems. For instance, the CinCECGTorso dataset has 1640 features and 4 classes. Future work will aim to develop methodologies that are effective for datasets with many features and multi-class classification problems to address these limitations.
Comments 5: The conclusions section (lines 393-406) summarises the main findings of the study and points out the effectiveness of data augmentation and DSP techniques in classifying time series data. The conclusions appear to be supported by the results, but it is recommended that the authors further discuss the implications of the results and contributions to existing research.
Response 5: Thank you for your comment. The following corrections have been made.
- Conclusions
In this paper, we proposed a new methodology that combines data augmentation and digital signal processing (DSP) techniques to enhance the performance of time series data classification. For data augmentation, we employed a noise injection technique to increase the diversity of the training data, augmenting each data sample tenfold by setting the Gaussian noise level to 30% of the standard deviation. Additionally, DSP was conducted through three stages—sampling, quantization, and Fourier transform—to extract important frequency features of the time series data. This approach improved the quality of the training data and maximized the model's generalization performance.
The experimental results showed that the proposed methodology significantly improved time series data classification performance compared to existing methods, demonstrating superior performance across various evaluation metrics. These findings confirm that the combination of data augmentation and DSP techniques is an effective tool for addressing time series data classification problems.
The results of this paper have the following implications. First, the integration of data augmentation and DSP techniques supports the effective learning of complex patterns in time series data, indicating that it can overcome limitations caused by data imbalance and noise. Second, achieving high classification performance with a small amount of data suggests potential applicability in fields where data collection is challenging. Furthermore, this paper contributes to existing research as follows. While existing time series classification methods have mainly focused on individual techniques, our paper confirmed the synergistic effect of combining data augmentation and DSP. This presents a new approach in the field of time series data classification and contributes to indicating the direction for future research.
In future work, we plan to further explore the scalability and applicability of the proposed methodology. We aim to develop effective methods for problems with many features and multi-class classification, thereby constructing a generalized model applicable to various time series datasets. Additionally, we will proceed with research to further enhance model performance by combining more complex signal processing techniques and advanced data augmentation methods. Through this, we expect to make practical contributions to solving time series data classification problems.
Comments 6: Overall, the English language quality of the article is good and clear. However, further grammar and spelling proofreading may be required in some places to ensure professionalism.
Response 6: Thank you for your comment. We have reviewed the overall English language in the paper, corrected grammatical and spelling errors, and refined the expressions to be clearer and more professional. This has improved the quality and readability of the manuscript.
Comments 7: The study presents a new approach that combines noise injection and DSP techniques, which is novel in the field of time series classification. However, it is recommended that the authors more clearly emphasise the uniqueness of their approach compared to existing techniques.
Response 7: Thank you for your comment. In response to your advice, we have revised the introduction and conclusion sections of our paper to more clearly emphasize the uniqueness of our approach compared to existing technologies. We have strengthened the comparisons with related studies and provided detailed explanations of how our method differentiates itself from existing noise injection and DSP techniques. This allows readers to clearly understand the innovation of the proposed approach and its contributions relative to existing research. The following corrections have been made.
- Introduction
Time series data play an important role in various fields such as finance [1], healthcare [2], and industrial processes [3], and accurate analysis of such data is essential for decision-making and performance improvement in these areas. However, due to the complexity and variability of time series data, effectively classifying them remains a challenging task. Time series data are characterized by nonlinearity and non-stationarity over time, and failure to properly capture these characteristics can lead to degraded classification performance. Existing time series classification methods include traditional statistical techniques [4], machine learning algorithms [5], and deep learning-based models [6], but they have several limitations due to data imbalance, noise, and limited data quantities [7]. For example, traditional methods struggle to capture complex patterns in the data, and deep learning models require large amounts of labeled data and high computational resources. Therefore, there is a need to develop new classification methodologies that can effectively handle the complexity of time series data and apply them to real-world problems across various domains. In this paper, we propose a novel approach that combines noise injection and digital signal processing techniques to overcome these limitations. Through this method, we enhance the features of the data and demonstrate that high classification performance can be achieved even with small amounts of data.
To address these issues, data augmentation techniques and DSP methods have emerged as useful tools [8]. Data augmentation techniques enhance data diversity by applying various transformations to existing data, thereby improving the generalization performance of models. Notably, noise injection is a technique that generates new data by adding random noise to the original data, significantly increasing data diversity. This allows the model to learn from a wider range of scenarios, thereby improving classification performance [9]. DSP is a method that transforms the features of time series data, enabling more effective extraction and representation of information. Through DSP techniques such as frequency analysis, filtering, and transformation, important features of time series data can be highlighted, or noise can be removed. This enhancement enables classification models to perform better [10].
Therefore, this paper aims to enhance the classification performance of time series data by combining data augmentation through noise injection and DSP methods. This approach enables the model to learn from the diverse variability of time series data and improves classification performance by emphasizing important features. The proposed methodology consists of three main steps. First, data augmentation is performed through noise injection. Second, DSP techniques are applied to the augmented data to extract important features from the time series data. Finally, the transformed features are classified using existing time series data models. This process aims to effectively model the complex patterns and dynamic changes of time series data, achieving high classification accuracy. By providing a detailed methodology, experimental results, and insights gained from this novel approach, this paper comprehensively evaluates the impact of integrating deep learning and DSP techniques on the field of time series data analysis.
The contributions of the proposed method in this paper are as follows:
- Novel Integration of Noise Injection and DSP for Time Series Classification: We propose a unique methodology that synergistically combines noise injection and digital signal processing techniques specifically for time series classification. This integrated approach is novel, as previous works have explored these techniques individually but not in conjunction.
- Enhanced Feature Extraction Framework: By applying DSP techniques—including sampling, quantization, and Fourier transform—we develop an effective framework for extracting salient frequency features from time series data. This improves feature quality and contributes to better model learning and classification performance.
- Improved Classification Performance with Limited Data: Our method demonstrates that high classification accuracy can be achieved even with limited datasets by increasing data diversity through controlled noise injection. This addresses a common limitation in time series analysis where data collection is challenging.
- Validation Across Diverse Domains: We validate the generalizability of our approach through experiments on various datasets from the UCR Time Series Classification Archive, covering multiple domains such as biology, healthcare, finance, and industry. This confirms that our method is not limited to a specific dataset or field.
The structure of this paper is as follows. Section 2 describes Time Series Data Augmentation and DSP. Section 3 details Extraction of Feature for Time Series Classification using Noise Injection. Section 4 presents the results and performance evaluation, and Section 5 concludes the paper.
- Yuan, A.E., Shoi, W.A rigorous and versatile statistical test for correlations between stationary time series. PLoS Biol. 2024, 22, e3002758.
- Masini, R.P., Medeiros, M.C., Mendes, E. F. Machine learning advances for time series forecasting. Econ. Surv. 2023, 37, 76-111.
- Fawa, H.I., Forestier, G., Weber, J., Idoumghar, L., Muller, P.A. Deep learning for time series classification: a review. Data Min Knowl Disco. 2019, 33, 917-963.
- Dhar, M., Dickinson, J.A., Berg, M.A. Efficient, nonparametric removal of noise and recovery of probability distributions from time series using nonlinear-correlation functions: Additive noise. Chem. Phys. 2023, 159, 054110.
- Conclusions
In this paper, we proposed a new methodology that combines data augmentation and digital signal processing techniques to enhance the performance of time series data classification. For data augmentation, we employed a noise injection technique to increase the diversity of the training data, augmenting each data sample tenfold by setting the Gaussian noise level to 30% of the standard deviation. Additionally, DSP was conducted through three stages—sampling, quantization, and Fourier transform—to extract important frequency features of the time series data. This approach improved the quality of the training data and maximized the model's generalization performance.
The experimental results showed that the proposed methodology significantly improved time series data classification performance compared to existing methods, demonstrating superior performance across various evaluation metrics. These findings confirm that the combination of data augmentation and DSP techniques is an effective tool for addressing time series data classification problems.
The results of this paper have the following implications. First, the integration of data augmentation and DSP techniques supports the effective learning of complex patterns in time series data, indicating that it can overcome limitations caused by data imbalance and noise. Second, achieving high classification performance with a small amount of data suggests potential applicability in fields where data collection is challenging. Furthermore, this paper contributes to existing research as follows. While existing time series classification methods have mainly focused on individual techniques, our paper confirmed the synergistic effect of combining data augmentation and DSP. This presents a new approach in the field of time series data classification and contributes to indicating the direction for future research.
In future work, we plan to further explore the scalability and applicability of the proposed methodology. We aim to develop effective methods for problems with many features and multi-class classification, thereby constructing a generalized model applicable to various time series datasets. Additionally, we will proceed with research to further enhance model performance by combining more complex signal processing techniques and advanced data augmentation methods. Through this, we expect to make practical contributions to solving time series data classification problems.
Comments 8: This research is important for the field of time series data classification as it provides a new approach that may improve classification performance. This is potentially valuable for practical applications in areas such as finance, healthcare and industry.
Response 8: Thank you for your comment.
Comments 9: The research methodology and experimental design appear to be scientifically sound, but it is recommended that the authors provide more detailed information on experimental validation and dataset selection to enhance the credibility of the study.
Response 9: Thank you for your comment. We have addressed this point by revising the relevant sections to provide more detailed information on experimental validation and dataset selection, as outlined in Response 4 of Comment 4. Please refer to Response 4 for the updated content.
Comments 10: The study presents a new approach that has the potential to improve the classification performance of time series data. The article is clearly structured, the methodology section is detailed, and the results section provides adequate data support. These strengths indicate that the study is a valuable contribution.
Response 10: Thank you for your comment.

Reviewer 3 Report
Comments and Suggestions for Authors
The paper proposes a methodology for improving time series classification through noise injection and digital signal processing (DSP). By using data augmentation techniques, the paper enhances the diversity of training data and extracts significant frequency features using DSP. However, there are some issues in this article needed to be improved.
1. The contribution part of the article is not brief enough to highlight the main points.
2. The contributions listed in the paper are somewhat generic and do not clearly demonstrate the novelty of the approach. The use of noise injection and DSP for time series classification has been explored in previous works.
3. The claim that the method is generalizable across various time series datasets is not fully supported by the results, as the experiments focus on a limited set of data.
4. The description of the experimental setup (Section 4.1) lacks specific details regarding dataset characteristics and preprocessing steps.
5. The conclusion section briefly summarizes the findings but does not fully discuss the broader implications of the methodology.
Comments on the Quality of English LanguageThe paper's English is adequate for conveying the research, but improving sentence structure, avoiding repetition, using active voice, and addressing minor grammatical issues will significantly enhance readability and professionalism.
Author Response
Comments 1: The contribution part of the article is not brief enough to highlight the main points.
Response 1: Thank you for your comment. We have revised the introduction and contributions sections to reflect your feedback. The following corrections have been made.
- Introduction
Time series data play an important role in various fields such as finance [1], healthcare [2], and industrial processes [3], and accurate analysis of such data is essential for decision-making and performance improvement in these areas. However, due to the complexity and variability of time series data, effectively classifying them remains a challenging task. Time series data are characterized by nonlinearity and non-stationarity over time, and failure to properly capture these characteristics can lead to degraded classification performance. Existing time series classification methods include traditional statistical techniques [4], machine learning algorithms [5], and deep learning-based models [6], but they have several limitations due to data imbalance, noise, and limited data quantities [7]. For example, traditional methods struggle to capture complex patterns in the data, and deep learning models require large amounts of labeled data and high computational resources. Therefore, there is a need to develop new classification methodologies that can effectively handle the complexity of time series data and apply them to real-world problems across various domains. In this paper, we propose a novel approach that combines noise injection and digital signal processing techniques to overcome these limitations. Through this method, we enhance the features of the data and demonstrate that high classification performance can be achieved even with small amounts of data.
To address these issues, data augmentation techniques and DSP methods have emerged as useful tools [8]. Data augmentation techniques enhance data diversity by applying various transformations to existing data, thereby improving the generalization performance of models. Notably, noise injection is a technique that generates new data by adding random noise to the original data, significantly increasing data diversity. This allows the model to learn from a wider range of scenarios, thereby improving classification performance [9]. DSP is a method that transforms the features of time series data, enabling more effective extraction and representation of information. Through DSP techniques such as frequency analysis, filtering, and transformation, important features of time series data can be highlighted, or noise can be removed. This enhancement enables classification models to perform better [10].
Therefore, this paper aims to enhance the classification performance of time series data by combining data augmentation through noise injection and DSP methods. This approach enables the model to learn from the diverse variability of time series data and improves classification performance by emphasizing important features. The proposed methodology consists of three main steps. First, data augmentation is performed through noise injection. Second, DSP techniques are applied to the augmented data to extract important features from the time series data. Finally, the transformed features are classified using existing time series data models. This process aims to effectively model the complex patterns and dynamic changes of time series data, achieving high classification accuracy. By providing a detailed methodology, experimental results, and insights gained from this novel approach, this paper comprehensively evaluates the impact of integrating deep learning and DSP techniques on the field of time series data analysis.
The contributions of the proposed method in this paper are as follows:
- Novel Integration of Noise Injection and DSP for Time Series Classification: We propose a unique methodology that synergistically combines noise injection and digital signal processing techniques specifically for time series classification. This integrated approach is novel, as previous works have explored these techniques individually but not in conjunction.
- Enhanced Feature Extraction Framework: By applying DSP techniques—including sampling, quantization, and Fourier transform—we develop an effective framework for extracting salient frequency features from time series data. This improves feature quality and contributes to better model learning and classification performance.
- Improved Classification Performance with Limited Data: Our method demonstrates that high classification accuracy can be achieved even with limited datasets by increasing data diversity through controlled noise injection. This addresses a common limitation in time series analysis where data collection is challenging.
- Validation Across Diverse Domains: We validate the generalizability of our approach through experiments on various datasets from the UCR Time Series Classification Archive, covering multiple domains such as biology, healthcare, finance, and industry. This confirms that our method is not limited to a specific dataset or field.
The structure of this paper is as follows. Section 2 describes Time Series Data Augmentation and DSP. Section 3 details Extraction of Feature for Time Series Classification using Noise Injection. Section 4 presents the results and performance evaluation, and Section 5 concludes the paper.
- Yuan, A.E., Shoi, W.A rigorous and versatile statistical test for correlations between stationary time series. PLoS Biol. 2024, 22, e3002758.
- Masini, R.P., Medeiros, M.C., Mendes, E. F. Machine learning advances for time series forecasting. Econ. Surv. 2023, 37, 76-111.
- Fawa, H.I., Forestier, G., Weber, J., Idoumghar, L., Muller, P.A. Deep learning for time series classification: a review. Data Min Knowl Disco. 2019, 33, 917-963.
- Dhar, M., Dickinson, J.A., Berg, M.A. Efficient, nonparametric removal of noise and recovery of probability distributions from time series using nonlinear-correlation functions: Additive noise. Chem. Phys. 2023, 159, 054110.
Comments 2: The contributions listed in the paper are somewhat generic and do not clearly demonstrate the novelty of the approach. The use of noise injection and DSP for time series classification has been explored in previous works.
Response 2: Thank you for your comment. In response, we have revised the introduction and contributions sections to be more concise and to demonstrate the novelty of our approach compared to existing methods. The revisions are detailed in Response 1.
Comments 3: The claim that the method is generalizable across various time series datasets is not fully supported by the results, as the experiments focus on a limited set of data.
Response 3: Thank you for your comment. We conducted experiments using various datasets from the UCR Time Series Classification Archive to verify the generalization performance of the proposed method. The UCR dataset is a widely used standard benchmark in time series classification research, consisting of 128 datasets spanning diverse domains such as biology, healthcare, finance, and industry. In our paper, we selected datasets from seven major categories within these domains to perform our experiments. Through this approach, we aimed to validate the effectiveness of the proposed method across multiple fields, not limited to a specific domain. Therefore, our experiments were not focused on a limited dataset but were designed to evaluate generalization performance across various domains. Additionally, we have revised the paper to include detailed descriptions of the datasets and preprocessing steps, allowing readers to clearly understand our experimental setup. The modifications have been detailed in Response 4, so please refer to it for further information.
Comments 4: The description of the experimental setup (Section 4.1) lacks specific details regarding dataset characteristics and preprocessing steps.
Response 4: Thank you for your comment. The following corrections have been made.
4.1. Dataset and Performance Evaluation Metrices for Experiments
In this section, the dataset and data augmentation techniques are described. All experiments are conducted on a system equipped with an Intel® Xeon® Silver 2.40 GHz CPU, 128GB RAM, and a single NVIDIA RTX 3090 GPU.
The performance evaluation in this paper is conducted using the UCRArchive2018 dataset. The UCR is a comprehensive collection of time series datasets used for analysis and classification, encompassing various domains. This dataset is widely employed for comparing and validating the performance of classification algorithms. One of the key features of the UCR dataset is that it includes dozens of datasets, each reflecting the characteristics of different applications and data types. This diversity is highly beneficial for evaluating the generalization ability of algorithms. However, due to the vast number of datasets, making efficient selections for experimentation is essential. In this paper, the following seven major categories are used to effectively categorize and select datasets from the UCR repository: Image Outline, Sensor Readings, Motion Capture, Spectrographs, Electric Devices, ECG, and Simulated. By selecting datasets that fit each of these categories, the experimental data can be utilized more efficiently for the intended purpose of the study. Categorizing the diverse UCR datasets in this manner allows for the evaluation of how consistently each algorithm performs across time series data with different characteristics. The datasets chosen from each category are split into training and testing sets, with 20% of the training data used for validation during the training process. All experiments are conducted using untrained models with the number of epochs fixed at 300.
Table 1. Proposed Methods Performance Evaluation.
|
Category |
Data |
LSTM |
|
|
Normal |
Ours |
||
|
Image Outline |
Crop |
63.26% |
72.80% |
|
ShapesAll |
69.50% |
72.17% |
|
|
InlineSkate |
30.55% |
33.09% |
|
|
Sensor readings |
DodgerLoopGame |
52.17% |
52.17% |
|
DodgerLoopWeekend |
73.91% |
73.91% |
|
|
FordA |
78.33% |
94.24% |
|
|
Motion capture |
Wafer |
99.03% |
99.48% |
|
CricketX |
44.87% |
54.87% |
|
|
Spectrographs |
Trace |
63.0% |
88.0% |
|
ChlorineConcentration |
58.80% |
73.59% |
|
|
Electric devices |
SonyAIBORobotSurface1 |
80.03% |
83.86% |
|
SonyAIBORobotSurface2 |
82.69% |
85.67% |
|
Table 1's "Normal" represents the performance evaluation of LSTM on each UCR dataset without any preprocessing. The results in Table 1. indicate that our methodology achieves higher performance across various datasets. This demonstrates that noise injection and DSP are effective for time series data classification. Specifically, analysis of the datasets with the most significant performance improvements reveals that our method shows substantial gains in few-shot data and binary classification problems. This finding substantiates the effectiveness of the proposed methodology for few-shot data and binary classification tasks. Conversely, the analysis of datasets where performance did not improve or even decline shows that these typically involve a high number of features and multi-class classification problems. For instance, the CinCECGTorso dataset has 1640 features and 4 classes. Future work will aim to develop methodologies that are effective for datasets with many features and multi-class classification problems to address these limitations.
Comments 5: The conclusion section briefly summarizes the findings but does not fully discuss the broader implications of the methodology.
Response 5: Thank you for your comment. The following corrections have been made.
- Conclusions
In this paper, we proposed a new methodology that combines data augmentation and digital signal processing techniques to enhance the performance of time series data classification. For data augmentation, we employed a noise injection technique to increase the diversity of the training data, augmenting each data sample tenfold by setting the Gaussian noise level to 30% of the standard deviation. Additionally, DSP was conducted through three stages—sampling, quantization, and Fourier transform—to extract important frequency features of the time series data. This approach improved the quality of the training data and maximized the model's generalization performance.
The experimental results showed that the proposed methodology significantly improved time series data classification performance compared to existing methods, demonstrating superior performance across various evaluation metrics. These findings confirm that the combination of data augmentation and DSP techniques is an effective tool for addressing time series data classification problems.
The results of this paper have the following implications. First, the integration of data augmentation and DSP techniques supports the effective learning of complex patterns in time series data, indicating that it can overcome limitations caused by data imbalance and noise. Second, achieving high classification performance with a small amount of data suggests potential applicability in fields where data collection is challenging. Furthermore, this paper contributes to existing research as follows. While existing time series classification methods have mainly focused on individual techniques, our paper confirmed the synergistic effect of combining data augmentation and DSP. This presents a new approach in the field of time series data classification and contributes to indicating the direction for future research.
In future work, we plan to further explore the scalability and applicability of the proposed methodology. We aim to develop effective methods for problems with many features and multi-class classification, thereby constructing a generalized model applicable to various time series datasets. Additionally, we will proceed with research to further enhance model performance by combining more complex signal processing techniques and advanced data augmentation methods. Through this, we expect to make practical contributions to solving time series data classification problems.
- Response to Comments on the Quality of English Language
Point 1: The paper's English is adequate for conveying the research, but improving sentence structure, avoiding repetition, using active voice, and addressing minor grammatical issues will significantly enhance readability and professionalism.
Response 1: Thank you for your comment. We have reviewed the overall English language in the paper, corrected grammatical and spelling errors, and refined the expressions to be clearer and more professional. This has improved the quality and readability of the manuscript.

Round 2
Reviewer 1 Report
Comments and Suggestions for Authors
After taken my comments into account, I think the paper is much better from now, especially the noise ratio.
I would like to thank the authors for their efforts to consider all my comments and modify the paper accordingly. Now, I think the paper is good enough for publishing with Sensors. I suggest the the paper can be accepted from this form.
Comments on the Quality of English LanguageGood enough
Reviewer 3 Report
Comments and Suggestions for Authors
The revisions have been made to address my previous comments. It is well written now. I have no further comments.